# Multidimensional scaling methods can reconstruct genomic DNA loops using Hi-C data properties

**Ryo Ishibashi**[ID]*

Department of Physics, Chuo University, Tokyo, Japan

* ryo.i.0609@outlook.jp

## Abstract

This paper proposes multidimensional scaling (MDS) applied to high-throughput chromosome conformation capture (Hi-C) data on genomic interactions to visualize DNA loops. Currently, the mechanisms underlying the regulation of gene expression are poorly understood, and where and when DNA loops are formed remains undetermined. Previous studies have focused on reproducing the entire three-dimensional structure of chromatin; however, identifying DNA loops using these data is time-consuming and difficult. MDS is an unsupervised method for reconstructing the original coordinates from a distance matrix. Here, MDS was applied to high-throughput chromosome conformation capture (Hi-C) data on genomic interactions to visualize DNA loops. Hi-C data were converted to distances by taking the inverse to reproduce loops via MDS, and the missing values were set to zero. Using the converted data, MDS was applied to the log-transformed genomic coordinate distances and this process successfully reproduced the DNA loops in the given structure. Consequently, the reconstructed DNA loops revealed significantly more DNA-transcription factor interactions involved in DNA loop formation than those obtained from previously applied methods. Furthermore, the reconstructed DNA loops were significantly consistent with chromatin immunoprecipitation followed by sequencing (ChIP-seq) peak positions. In conclusion, the proposed method is an improvement over previous methods for identifying DNA loops.

## 1 Introduction

Gene expression is regulated by one- and three-dimensional chromatin structures [1], which can vary between cells [2]. DNA loops are formed via interactions between proximal promoters and distal enhancers or insulators [3]. Although the effect of these DNA loops on gene expression is well established, the detailed mechanism underlying gene expression regulation is poorly understood. In particular, diseases caused by genetic mutations, such as cancer, are promoted by irregular chromatin structures [4, 5]. Determining when and where DNA loops form along with visualization development are vital for disease prevention. High-throughput chromosome conformation capture (Hi-C) examines genomic interactions by determining the number of genomic contacts [3]; however, it presents numerous problems. Hi-C data are noisy [6], with most of them representing the average of an entire cell population, as single-cell

**Competing interests:** The authors have declared that no competing interests exist.

Hi-C data are uncommon owing to the cost of the method. Compared to Hi-C contact frequency between initially close coordinates, the frequency between the distant coordinates that form DNA loops is typically underestimated [7].

These aspects complicate genome structure prediction. Previous studies have reproduced the entire 3D genome structure from noisy Hi-C data using two methods: 1) The first involves converting Hi-C contact frequencies into distance data using an arbitrary function and then applying the multidimensional scaling (MDS) or t-distributed stochastic neighbor embedding (t-SNE). The 3D genome structure is then reproduced by considering the Poisson distribution and other factors in the contact frequencies and using the Markov chain Monte Carlo method to ensure consistency. 2) The stoHi-C method reproduces the genomic 3D structure by applying t-SNE to yeast cell Hi-C data [8]. As a result, a genomic structure similar to the yeast cell chromosome map is obtained. In addition, the miniMDS method divides high-resolution Hi-C data into multiple parts and applies MDS to low-resolution data [9]. Then, by integrating these data, a robust chromosome structure is produced. However, these methods have only been applied to reproduce whole-genome 3D structures, and further biological verification has not yet been performed. In previous studies, the 3D structure was only verified by determining the correlation between the input (Hi-C contact frequency or transformed distance data) and the output (3D genome coordinates). Therefore, this study presents a method of reproducing genomic 3D structures solely based on DNA loop structure. Several methods, such as HiC-CUPs [10] and cLoops [11], have been proposed to directly call DNA loops instead of reconstructing the genome structures. However, the genome structure must be visualized to investigate the physical interactions of genomes that cannot be understood using raw Hi-C data or loop-only calling methods. In addition, the method proposed here is expected to improve our understanding on the relationship between the dynamic state of complex genomes and genomic functions, such as gene expression regulation.

Previously, the average number of Hi-C contact frequencies at the same base distance was replaced to account for missing Hi-C data. However, this approach is ineffective to represent the genome structure [7]. Here, I demonstrate that the chromatin structure can be reproduced by setting the missing values to zero. To the best of my knowledge, this is the first study to apply MDS to Hi-C data to reproduce DNA loop-specific genomic structures without determining missing values. DNA loop reproduction is robust and missing values can be disregarded by exploiting the ability of MDS to reproduce positions relative to that of Hi-C data. Compared to results of previous methods, this study findings revealed significantly more transcription factors involved in loop formation. The results are also significantly consistent with the ChIP-seq peaks and biological findings. The proposed method of reproducing DNA loops for Hi-C data, which vary based on the experimental specifications, is expected to provide a basis to elucidate the mechanisms underlying the transcription and organization of 3D chromatin structures.

## 2 Materials and methods

### 2.1 Data

Hi-C datasets measure the physical interactions of genomes. In this study, the following seven representatives Hi-C datasets were used (Table 1).

All data were retrieved from the Gene Expression Omnibus (GEO) database [12]. In this study, all .hic files were loaded into the R package "straw" [13] with vanilla coverage for normalization.

**2.1.1 GSE201353.** The GSE201353 database includes Hi-C data collected at eight time points with a resolution of 10,000 bp; the cells are quiescent macrophages derived from human

**Table 1. Information on the seven Hi-C datasets analyzed in this study.**

| Series | Cell type | Read depth | Bin size [bp] | Year | Instrument |
|---|---|---|---|---|---|
| GSE201353 | macrophages | - | 10,000 | 2022 | Illumina NovaSeq 6000 |
| GSE141067 | osteosarcoma | - | 50,000 | 2020 | Illumina NovaSeq 6000 |
| GSE149103 | pancreas | 3 billion | 10,000 | 2021 | HiSeq X Ten |
| GSE160235 | colorectal gland | 450 million | 10,000 | 2021 | Illumina NovaSeq 6000 |
| GSE167150 | breast | 600 million | 10,000 | 2022 | Illumina NovaSeq 6000 |
| GSE168470 | lymphoma | - | 10,000 | 2021 | Illumina HiSeq 2500 and Illumina NextSeq 500 |
| GSE143465 | renal | 3 billion | 10,000 | 2021 | Illumina NovaSeq 6000 |

THP-1 macrophages using Illumina NovaSeq 6000 (GSM6061759 to GSM6061798) [14] (Illumina Inc., San Diego, CA, USA). Reed et al. investigated the interrelationship between 3D chromatin structure and transcription [14]. These data were collected after treatment with LPS/IFNg for 0, 0.5, 1, 1.5, 2, 4, 6, and 24 h.

**2.1.2 GSE141067.** The GSE141067 database includes Hi-C data collected at eight time points with a resolution of 50,000 bp from human U2OS osteosarcoma cells [15]. Kang et al. investigated histone modifications and long-range chromosome interactions after mitosis [15]. These time series data were collected during the cell cycle (0 min (metaphase) and 35 min (anaphase/telophase); 60 min (cytokinesis); and 90, 120, 180, 240, and 360 min (G1) (GSM4194449 to GSM4194464)) using Illumina NovaSeq 6000.

**2.1.3 GSE149103.** The GSE149103 database includes Hi-C data with a resolution of 10,000 bp and three different pancreatic cells: immortalized cells (GSM4490488), PANC-1 (GSM4490510), and Capan-1 (GSM4490532), which were assessed using HiSeq X Ten [16]. Ren B et al. established that chromatin loops were significantly altered and associated with epigenetic changes in metastatic pancreatic cancer cells [16].

**2.1.4 GSE160235.** The GSE160235 database includes Hi-C data collected at three time points with a resolution of 10,000 bp using a colorectal adenocarcinoma cell line. The control, RNAPll-degron at the post-mitotic phase, G2 phase, and transition to G1 phase, were assessed using Illumina NovaSeq 6000 [17]. Zhang et al. concluded that RNA polymerase II is required for chromatin reorganization [17].

**2.1.5 GSE167150.** The GSE167150 database includes Hi-C data for breast cancer cell lines with a resolution of 10,000 bp. These data were obtained for six breast cancer subtypes, including triple-negative breast cancer (TNBC) and normal cells, using Illumina NovaSeq 6000 [18]. Kim et al. found that, compared to the other five breast cancer subtypes, TNBC has a more rapid progression, disrupted chromatin structure, and tissue-specific loops [18].

**2.1.6 GSE168470.** The GSE168470 database includes Hi-C data for lymphoma with a resolution of 10,000 bp. These data were obtained for lymphoma cancer subtypes, including WSU-DLCL2, DLBCL, and germinal center B-cells, using Illumina HiSeq 2500 and Illumina NextSeq 500 [19]. Sungalee et al. found that H3K27ac dynamics may regulate genomic interactions and maintain oncogene expression [19].

**2.1.7 GSE143465.** The GSE143465 database includes Hi-C data for renal cancer with a resolution of 10,000 bp. These data were obtained for renal cancer subtypes, including N-IDR, WT/A9, and N-IDR FS/A9, using Illumina NovaSeq 6000 [20]. Ahn et al. demonstrated that phase-separated NUP98-HOXA9 induces chromatin loops in a proto-oncogene [20].

## 2.2 Methods

Fig 1 shows the flowchart of analyses performed in this study.

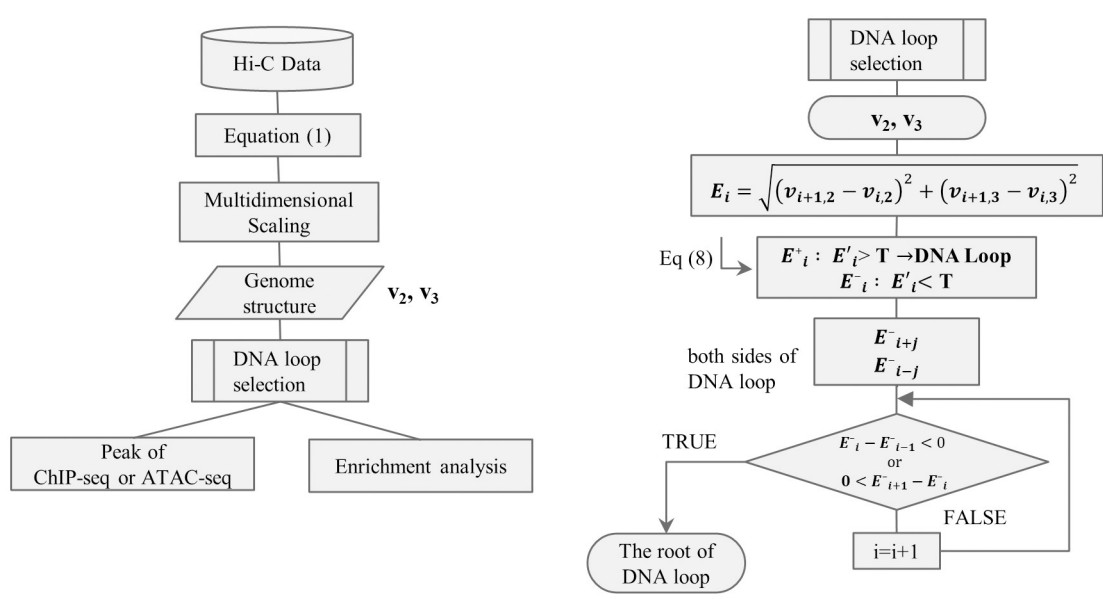

**Fig 1. The flowchart of analyses performed in this study.**

**2.2.1 Missing values in Hi-C data.** The missing values in the Hi-C datasets were set to zero to differentiate between DNA with and without loops.

**2.2.2 Preprocessing to clearly represent DNA loops.** Hi-C contact frequency between distant coordinates that form DNA loops are underestimated compared to Hi-C contact frequency between initially close coordinates [7]. Thus, the only important pre-processing step was to reduce the gap between Hi-C contact frequencies using the multiplying Eq (1).

$$
d'_{ij} = \begin{cases} d_{ij} & , \mid i - j \mid \le 5 \quad [\times 10,000 \quad \text{bp}] \\ d_{ij} \times \log_e \mid i - j \mid & , \quad \mid i - j \mid > 5 \quad [\times 10,000 \quad \text{bp}] \end{cases} \tag{1}
$$

where $i$ and $j$ are defined as the bin coordinates that divide the genome coordinates by the resolution and $d_{ij}$ represents the Hi-C contact frequency. The Hi-C data with a resolution of 10 kbp had a large default value for the number of contact frequencies below 50 kbp between the coordinates; therefore, 50 kbp was considered as the cutoff. The Eq (1) is a dependent function of nucleotide distance and causes more pronounced Hi-C contact frequency between distant coordinates that form DNA loops (Figs 2 and 3).

For Hi-C contact frequencies above 50 kbp, the gap in the Hi-C data was filled by multiplying by the natural logarithm of the distance. For instance, adjusting the cutoff value of 50–70 kbp did not significantly change the structure of the genome. However, setting the cutoff to 20 kbp altered the genome structure because a Hi-C contact frequency below 20 kbp will be low because of the natural logarithm. Here, setting the cutoff value to 50 kbp reproduces a genome structure that represents the DNA loop. However, there is no strict biological basis for this.

Various other functions were also considered. For instance, multiplying the number of Hi-C contact frequencies by a constant multiple (Eq (2)) and the original Hi-C data did not

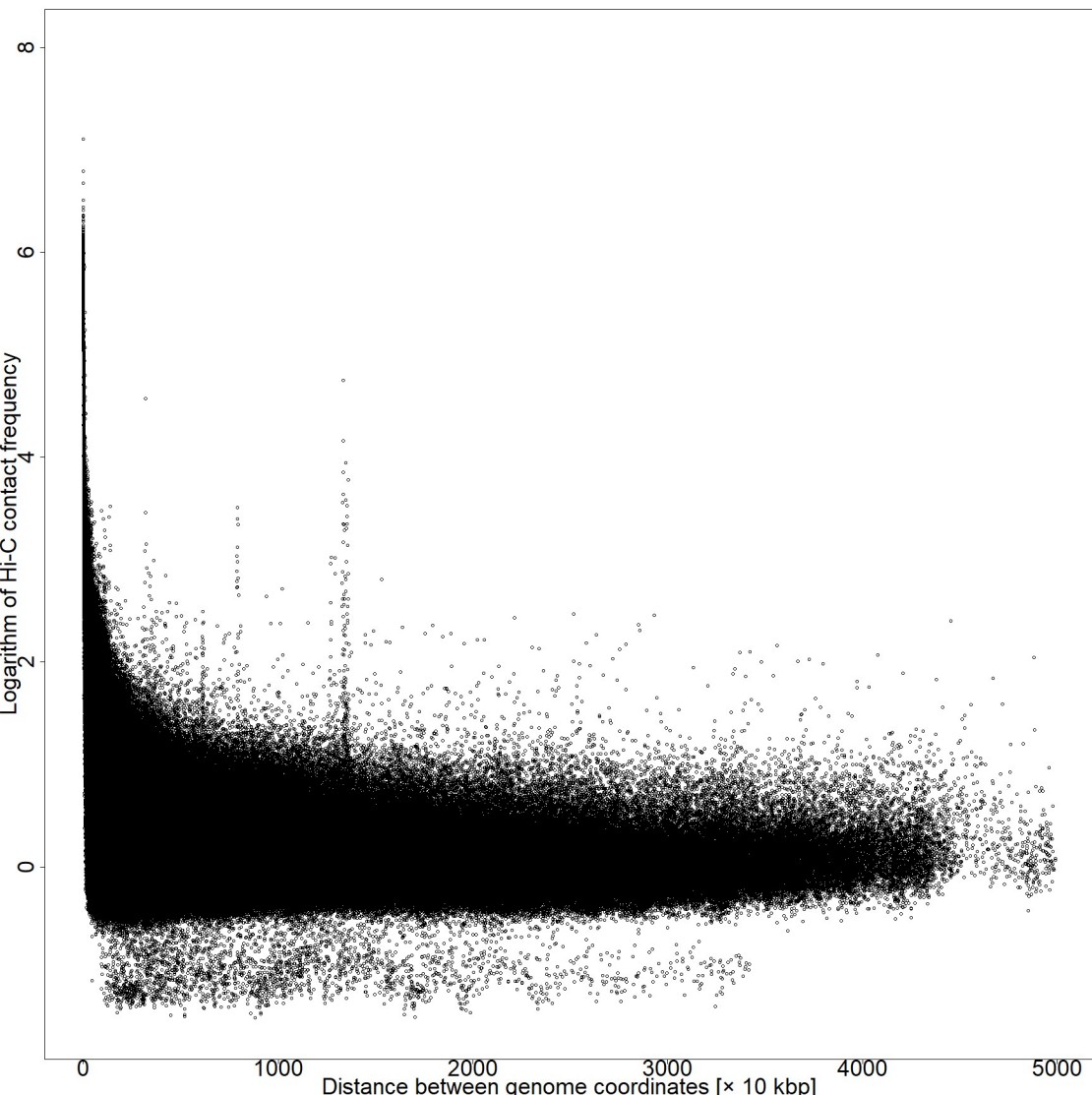

**Fig 2. Plot of the distance between coordinates versus the natural logarithm of Hi-C contact frequency before multiplying by weight 0–50,000 kbp of chromosome 5 by series GSM6061774.**

reproduce a genomic structure consistent with the biological findings (S1 File).

$$d'_{ij} = \begin{cases} d_{ij} & , \mid i-j \mid \leq 5 \quad [\times 10,000 \quad \text{bp}] \\ d_{ij} \times a \mid i-j \mid & , \quad \mid i-j \mid > 5 \quad [\times 10,000 \quad \text{bp}] \end{cases} \tag{2}$$

If $d'_{ij}$ now has a large value, then, the distance between genomic coordinates is small. Therefore, the reciprocal of $d'_{ij}$ was taken to treat the processed Hi-C contact frequency as a distance. However, the missing values are zero, and all $d'_{ij}$ values were added to 1. Then, the inverse of the $d'_{ij}$ was used as the distance between genome coordinates. Moreover, the diagonal component $d'_{ii}$ was set to zero because of the number of Hi-C contact frequencies for the same

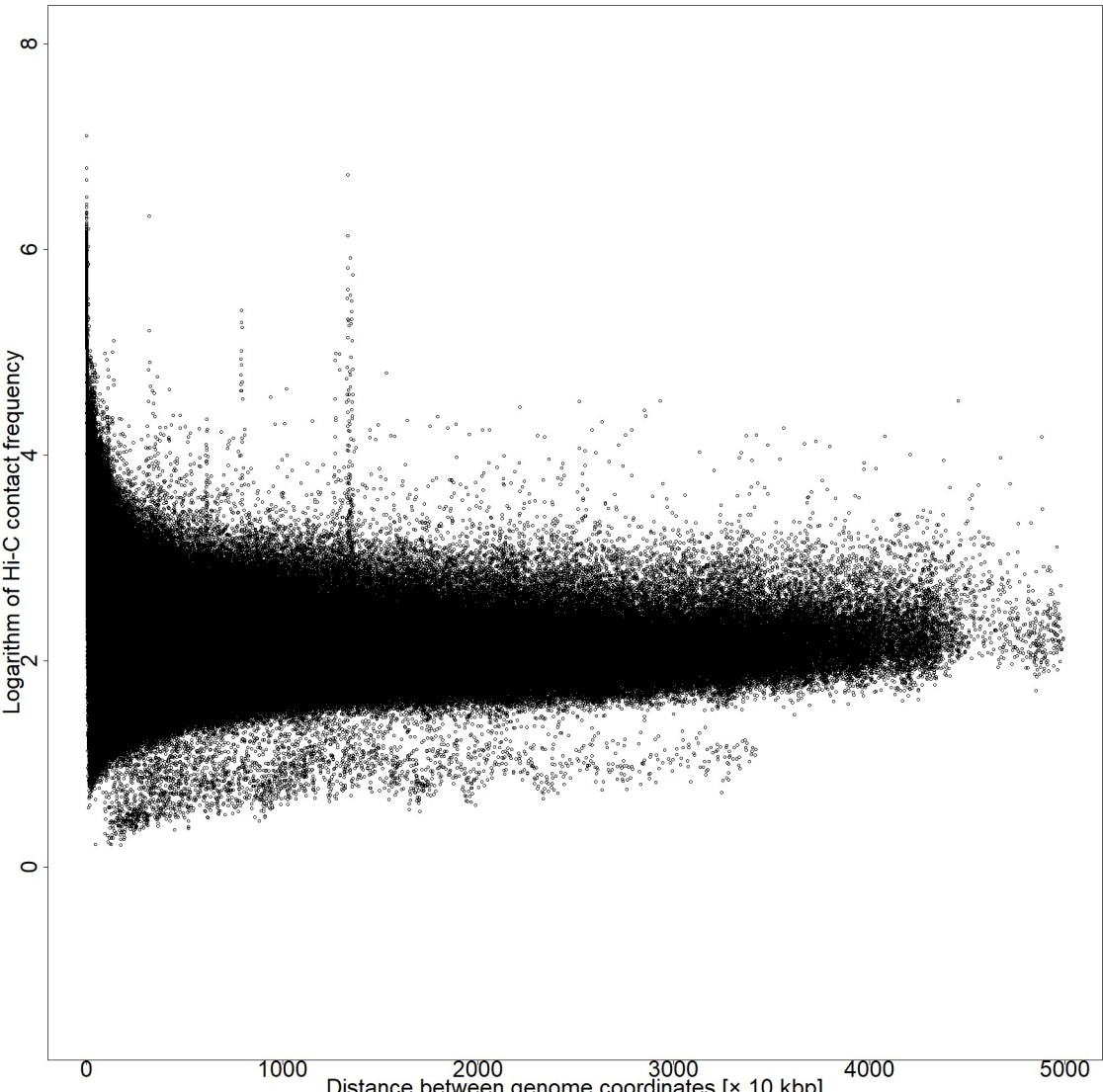

**Fig 3. Plot of the distance between coordinates versus the natural logarithm of Hi-C contact frequency after multiplying by weight 0–50,000 kbp of chromosome 5 by series GSM6061774.**

coordinate.

$$D_{ij} = \begin{cases} \dfrac{1}{d'_{ij} + 1} & , i \neq j \\ 0 & , i = j \end{cases} \qquad (3)$$

The 'PHi-C' method reproduces genome structures by considering the simple macromolecular model of the genome (a single chromosome) as a 'linked bead' [21]. PHi-C accurately estimates genome structure in four dimensions but requires dense Hi-C data. In this case, a versatile model was desired, hence the reciprocal frequency of interactions between two particles in the model (i.e., between two bins or restriction fragments in the genome) was adopted

as the distance. The inverse of the power of the frequency is debatable, but it was set to 1 as used in fractal globule models [22].

**2.2.3 Multidimensional scaling.** The MDS construction method finds coordinates in the original spatial data based on similarity or distance data [23]. Although Euclidean distance is generally used as proximity for similarity, weighted Euclidean distance, Manhattan distance, and Minkowski space are also commonly used. Hi-C data have a larger contact frequency when the genomic coordinate distances are closer; however, as MDS deals with distance data, Hi-C data are transformed.

The reciprocal of Hi-C data reveals that the closer the distance between genomic coordinates, the smaller the value. By contrast, the farther the distance, the larger the value. Furthermore, the diagonal component was assigned a value of zero.

The MDS algorithm is briefly described below. When the number of elements is $N$, the input is a square matrix $\boldsymbol{D}$, where $D_{ij}$ is the distance between the $i$-th and $j$-th genomic coordinates. Then, $x_n$ was defined as each genomic coordinate to be reproduced as MDS output. The original genome structure $\boldsymbol{X}$ is obtained from a distance $\boldsymbol{D}$ of the genome coordinates as follows. $\mathbf{D}^{(2)}$ was defined as the matrix of all components of the distance matrix $\mathbf{D}$ squared, and multiply $\mathbf{D}^{(2)}$ by the $N{\times}N$ centralization matrix $\boldsymbol{J}\left(= \boldsymbol{E} - \frac{1}{N}\mathbf{1}\right)$ from both sides.

$$
\begin{aligned}
-\frac{1}{2}\boldsymbol{J}\boldsymbol{D}^{(2)}\boldsymbol{J}^t &= -\frac{1}{2}\boldsymbol{J}\{\mathrm{diag}(\boldsymbol{X}\boldsymbol{X}^t)\mathbf{1} - 2\boldsymbol{X}\boldsymbol{X}^t + \mathbf{1}\mathrm{diag}(\boldsymbol{X}\boldsymbol{X}^t)\}\boldsymbol{J}^t \\
&= \boldsymbol{J}\boldsymbol{X}\boldsymbol{X}^t\boldsymbol{X}^t = \boldsymbol{J}\boldsymbol{X}(\boldsymbol{J}\boldsymbol{X})^t \\
&= \boldsymbol{X}^*\boldsymbol{X}^{*t} = \boldsymbol{K}
\end{aligned}
\tag{4}
$$

$\boldsymbol{X}^*$ is the centered data matrix and $\boldsymbol{K}$ is the inner product $N \times N$ matrix obtained from the centered data matrix. Next, consider a matrix $\boldsymbol{X}$ that satisfied the eigenvalue decomposition of $\boldsymbol{K}$ and $\boldsymbol{K}$ can be decomposed as follows:

$$
\boldsymbol{K} = \boldsymbol{V}^t\boldsymbol{A}\boldsymbol{V}
\tag{5}
$$

where $\boldsymbol{V}$ is an unitary matrix, and $\boldsymbol{A}$ is an eigenvalue matrix. Then, from Eqs (4) and (5),

$$
\boldsymbol{X} = \sqrt{\boldsymbol{A}}\boldsymbol{V}
\tag{6}
$$

Eq (6) can obtain the desired genomic structure $\boldsymbol{X}$. Eigenvalues and eigenvectors are arranged in order of magnitude of the eigenvalues. Empirically, the second and third eigenvectors $\boldsymbol{v_2}$, $\boldsymbol{v_3}$ after eigenvalue decomposition frequently retain their original structure. This was verified in my previous paper with a simple simulation [7]. Therefore, after eigenvalue decomposition, $\boldsymbol{v_2}$ and $\boldsymbol{v_3}$ of the eigenvector $\boldsymbol{V}$ were chosen as the genomic structure. Here, this structure is defined as a tentative chromosome structure (Fig 4).

**2.2.4 Criteria for DNA loop selection.** The criteria for selecting DNA loops from the tentative chromosomes are described below. Then, the Euclidean distance between $\boldsymbol{v_2}$ and $\boldsymbol{v_3}$ was calculated as follows:

$$
\boldsymbol{E_i} = \sqrt{\left(\boldsymbol{v_{i+1,2}} - \boldsymbol{v_{i,2}}\right)^2 + \left(\boldsymbol{v_{i+1,3}} - \boldsymbol{v_{i,3}}\right)^2}
\tag{7}
$$

where $i$ is the locus divided by the resolution and $\boldsymbol{E_i}$ is the distance of the genome at $i$ and $i+1$. The length of the loops is dependent on the cell but was averaged over 170–200 kb in quiescent macrophages [14]. Therefore, the average of ten points (100 kbp) each of $\boldsymbol{E_i}$ was calculated and used as the distance. Since this process reduced the total data by 100 kbp, insufficient data were represented by averages obtained at every 90, 80, 70, 60, and 50 kbp from both sides as

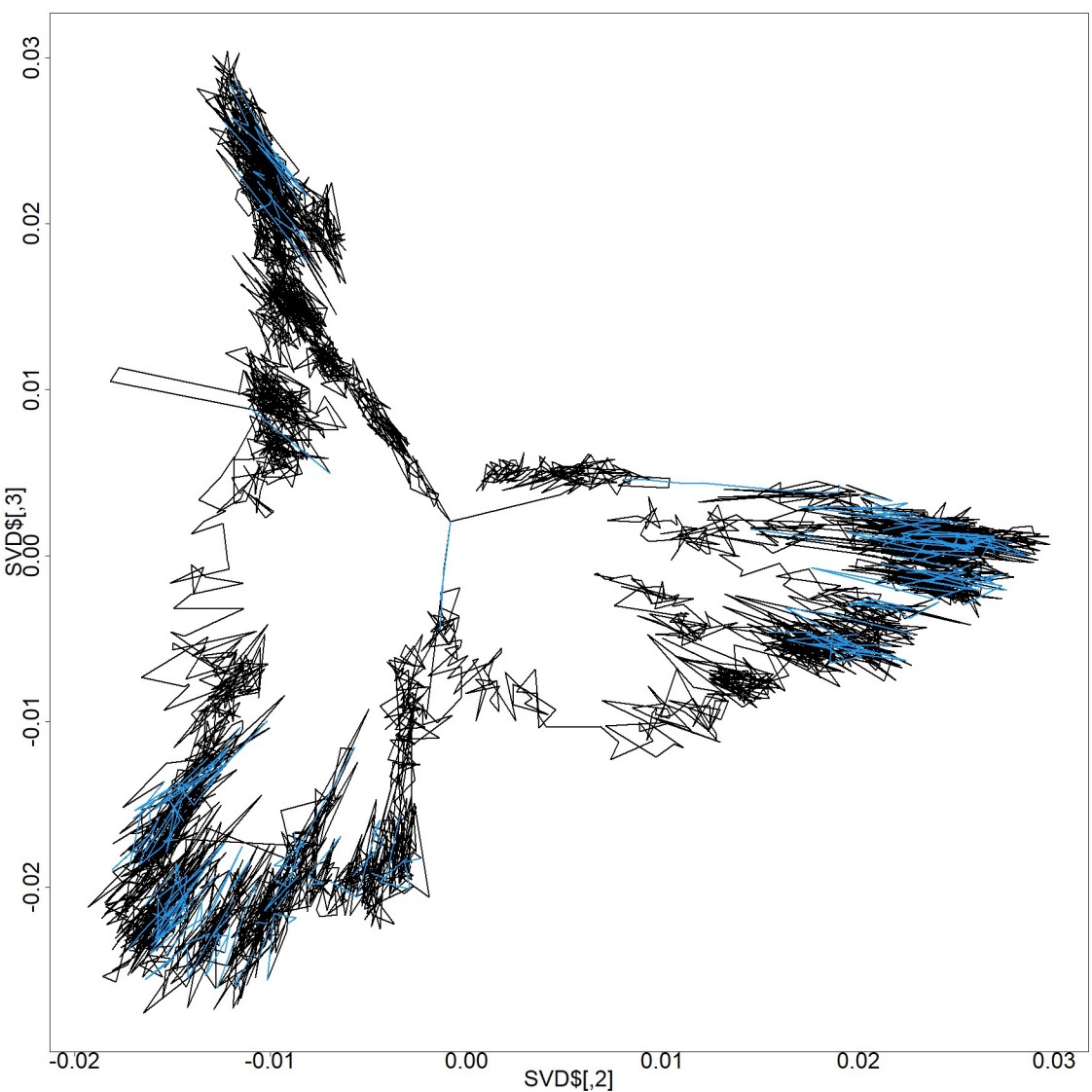

**Fig 4. Plot of the second and third eigenvectors.** Chromosome 5 0–50,000 kbp by series GSM6061774 after MDS. The blue line represents the root of the DNA loop, which is defined in Section 2.2.4.

follows.

$$
\mathbf{E}_i' =
\begin{cases}
\dfrac{\Sigma_{j=1}^{k} \mathbf{E}_{i+j}}{k} & , i = 0, k = 5, 6, \ldots, 9 \\[2ex]
\dfrac{\Sigma_{j=0}^{9} \mathbf{E}_{i+j}}{10} & , i = 1, 2, \ldots, N-1 \\[2ex]
\dfrac{\Sigma_{j=0}^{k} \mathbf{E}_{i+j}}{k+1} & , i = N-9, k = 8, 7, \ldots 4
\end{cases}
\tag{8}
$$

where $i$ represents the coordinates divided by the resolution and $N$ is the number of elements of $i$. In addition, $j$ and $k$ are elements to complement the reduced $i$ by taking the average of $\mathbf{E}_i$. This smoothing was performed at 100 kbp, which is neither too long nor too short. Further

investigation is needed when dealing with other resolutions. The criteria for obtaining coordinates as DNA loops are shown below. The threshold value was set at two times the average distance between genomic coordinates $E$. The coordinates above the threshold were considered the regions that form DNA loops.

However, as enhancers and promoters are located at the ends of DNA loops, subthreshold coordinates were also evaluated. Therefore, in this study, the following criteria were established (Fig 1):

Formulaically, let T be the threshold, $E_i'$ greater than the threshold $T$ is denoted as $E_i^+$ and is a DNA loop, where $i$ is the coordinates divided by the Hi-C resolution. In contrast, $E_i'$ smaller than the threshold value is denoted as $E_i^-$. When proceeding from both sides of the DNA loop $E_i^+$ to directions $i + j$ and $i - j$ ($j$ is any natural number) that are smaller than the threshold value $T$, the root of the DNA loop is defined as $E_i^- - E_{i-1}^- < 0$ and $0 < E_{i+1}^- - E_i^-$. Especially, the location of the local minimum $E_i^-$ was set as the end of the DNA loop.

Therefore, the coordinates of the blue points in Fig 5 are considered enhancer and promoter regions.

**2.2.5 Enrichment analysis.** Genes in the DNA loops selected by the MDS-based method were obtained using the R package "BiomaRt" [24]. These were analyzed using the enrichment analysis software "g:profiler" [25]. Enrichment analyses evaluate the significance of the gene list in the selected region.

**2.2.6 Consistency between DNA loops and ChIP-seq or ATAC-seq peaks.** Chromatin immunoprecipitation followed by sequencing (ChIP-seq) is an experimental technique for mapping DNA-binding proteins and histone modifications [26]. Assay for transposase-accessible chromatin with high-throughput sequencing (ATAC-seq) investigates DNA accessibility using hyperactive Tn5 transposase [27]. ATAC-seq helps identify open chromatin regions and assesses transcription factor occupancy. DNA loops have many arbitrary protein interventions to promote or regulate transcription, and three spatial spaciousness, as such, selected DNA loops and ChIP-seq and ATAC-seq should match the peaks in the data. Therefore, the DNA loop regions selected by the MDS-based method were rechecked to determine whether they matched the locations of ChIP-seq or ATAC-seq peaks. When a partial match was found between the position of the top 300 peaks and the DNA loop regions selected by the MDS-based method, the loop was counted. Then, Fisher's exact probability test was performed to determine whether the selected DNA loops significantly matched the peaks compared to randomly selected loops.

ChIP-seq data for H3K27ac, H3K27me3, and CTCF were analyzed; where H3K27ac is an active enhancer marker, CTCF is involved in DNA loops, and H3K27me3 is involved in heterochromatin but is also found to be abundant in DNA loops and represses gene expression [28].

## 3 Results

Eqs (1) and (3) were applied to the Hi-C data to obtain a genome structure that differentiated between the DNA with and without loops. Then, DNA loops obtained through the MDS-based method were selected using Eqs (7) and (8) (S1 File). Genes in the DNA loops were obtained using the R package "BiomaRt" [24]. The number of DNA loops obtained by the MDS-based method and genes observed in the selected region in GSM6061774 to GSM6061778 (THP-1 90 min biological replicates 1 to 5) are summarized in Table 2.

Salzberg et al. mined four databases and estimated that the number of human genes was between 19,901 and 21,306 [29]. Therefore, 21,306 genes were selected for analysis.

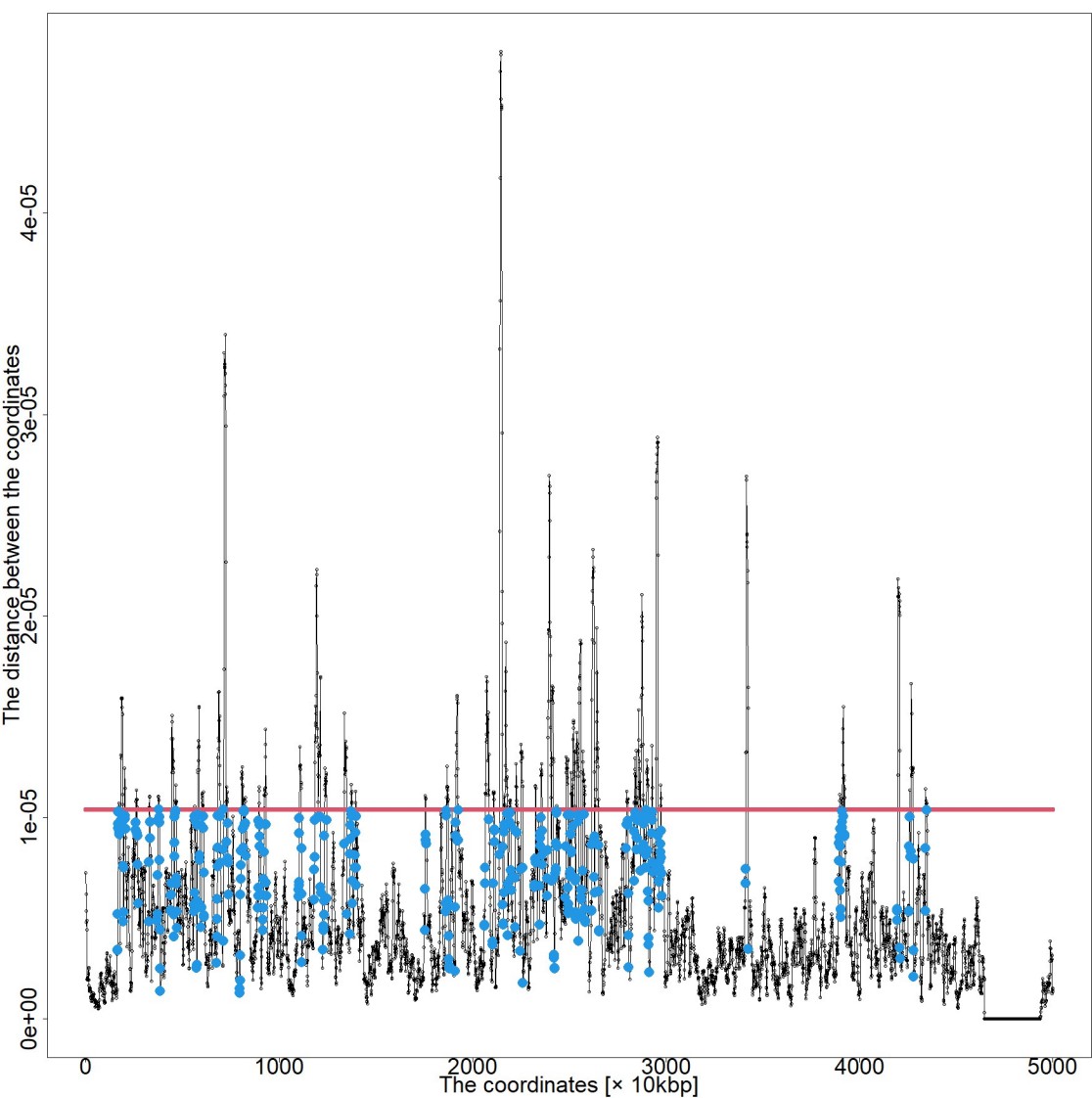

**Fig 5. Distance plot between coordinates of chromosome 5 0–50,000 kbp by series GSM6061774.** The red line is the threshold, and the blue points are the roots of the DNA loop, which is defined in Section 2.2.4.

Independent of the replication, the number of DNA loops is nearly 10%, and the number of genes is nearly 22%. Genes account for only approximately 1.5% of the DNA in the human genome. Therefore, DNA loop regions selected based on the MDS method should be located in gene-dense regions.

**Table 2. Proportion of DNA loop regions selected by the multidimensional scaling (MDS)-based method and genes in the DNA loops.**

| Data | DNA loop regions [×10, 000$bp$] | Gene |
|---|---|---|
| GSM6061774 | 31860/318380 = 0.100 | 5002/21306 = 0.235 |
| GSM6061775 | 30835/318380 = 0.097 | 4819/21306 = 0.226 |
| GSM6061776 | 31638/318380 = 0.099 | 4873/21306 = 0.229 |
| GSM6061777 | 30581/318380 = 0.096 | 4864/21306 = 0.228 |
| GSM6061778 | 32555/318380 = 0.102 | 4560/21306 = 0.214 |

## 3.1 Enrichment analysis

All genes containing DNA loops by the MDS-based method were subjected to enrichment analysis using "g:profiler" [25].

**3.1.1 GSE201353 (macrophages).** Transcription factors related to DNA loop formation were enriched for data collected at all eight time points and five biological replicates. Enrichment analysis for the data collected at 90 min, when DNA loop formation increases rapidly, are shown in Table 3.

HOXA is a family of homeobox genes. HOXA13, HOXA1, HOXA5, HOXA6, HOXA7, HOXA9, and HOXA10 are involved in prostate cancer, and they form a DNA loop with a locus that induces prostate cancer and regulates gene transcription [30]. CCAAT-enhancer-binding proteins (CEBPs) promote specific gene expression in many organ cells, including hepatocytes, hematopoietic cells, and kidney cells, through interactions with promoters. These proteins collect core activators and open chromatin structures [31]. CEBPA is a member of the CEBPs and interacts with promoters to promote the expression of specific genes. It mobilizes core activators, opens chromatin structures, and carries general transcription factors [32]. SATB1 (SATB Homeobox 1) is known as a global chromatin organizer. SATB1 forms DNA loops by linking the nuclear matrix regions (MARs) to the nuclear matrix at fixed distances [33]. The high mobility group protein, HMGIY is involved in transcription and replication and coordinates the nucleosome and chromatin structure [34, 35]. HMGIY can bend straight DNA and for DNA loops in Cos-7 cells [36].

**3.1.2 GSE141139 (osteosarcoma).** Genes in regions that exceeded the data threshold were also used in the enrichment analysis. Transcription factors involved in DNA loop formation were enriched in all data collected at eight time points. Enrichment analysis for data collected at 90 min, when long-range interactions were completed, are listed in Table 4.

Estrogen receptor alpha (ER$\alpha$), one of the two main types of estrogen receptors, is a nuclear receptor activated by ER that is involved in transcription activation and DNA binding. This receptor has been clinically implicated in breast, ovarian, and other types of cancers. Distal ER$\alpha$BS interacts with proximal sites to form chromatin loops in human breast adenocarcinoma cells [37]. Retinoic acid receptor alpha (RAR$\alpha$) regulates transcription in a ligand-dependent manner. Related diseases include acute myeloid leukemia and leukemia [38]. Phosphorylated RAR$\alpha$ is located in the R1 and R2 regions of the Cyp26A1 promoter and mobilizes RNA polymerase and TFIIH to form a DNA loop [39]. Histone deacetylase 1 (HDAC1) plays a vital role in regulating eukaryotic gene expression, and mobilization of

**Table 3. Enrichment analysis of Hi-C data collected at 90 min by g:Profiler.**

| Term_Name | Term_ID | Adjusted_p_value |
|---|---|---|
| HOXA13; motif: ATAAMA; match class: 1 | TF:M01292_1 | $2.81 \times 10^{-9}$ |
| SATB1; motif: NTTTAT; match class: 1 | TF:M03564_1 | $4.03 \times 10^{-9}$ |
| HOXA7; motif: GYMATTAN | TF:M10690 | $2.55 \times 10^{-7}$ |
| C/EBPgamma; motif: YTBATTTCARAAW; match class: 1 | TF:M00622_1 | $1.03 \times 10^{-6}$ |
| HOXA10; motif: NNNGYAATAAAATNNW | TF:M01464 | $1.68 \times 10^{-6}$ |
| HMGIY; motif: NNKKNAWTTTNYTNN; match class: 1 | TF:M01010_1 | $1.25 \times 10^{-4}$ |
| TBP; motif: NGNNTATAAAA | TF:M03581 | $8.14 \times 10^{-4}$ |
| IRF-4; motif: GAAARTA; match class: 1 | TF:M01883_1 | $1.63 \times 10^{-3}$ |
| C/EBPalpha; motif: NRTTGTGCAAYNN | TF:M09596 | $3.27 \times 10^{-3}$ |
| C/EBP; motif: NNATTGCNNAANNN; match class: 1 | TF:M00190_1 | $3.89 \times 10^{-3}$ |

**Table 4. Enrichment analysis of Hi-C data collected at 90 min by g:Profiler.**

| Term_Name | Term_ID | Adjusted_p_value |
|---|---|---|
| NR1B1; motif: NRGGNCRTGACCTN; match class: 1 | TF:M11796_1 | $3.05 \times 10^{-10}$ |
| AP-2gamma; motif: NTGSCCTGRGGSNN | TF:M09591 | $3.98 \times 10^{-7}$ |
| HDAC1; motif: KGCARGGTC; match class: 1 | TF:M07041_1 | $6.75 \times 10^{-7}$ |
| GATAD2A; motif: CCTKTG; match class: 1 | TF:M09726_1 | $2.89 \times 10^{-5}$ |
| Sp3; motif: NGCCACGCCCMCN; match class: 1 | TF:M12153_1 | $3.93 \times 10^{-5}$ |
| BTEB4; motif: NCCACGCCCM; match class: 1 | TF:M12186_1 | $8.84 \times 10^{-5}$ |
| TBX2; motif: NNNGTGTSNN; match class: 1 | TF:M12026_1 | $2.75 \times 10^{-4}$ |
| NF1C; motif: WGCCARR; match class: 1 | TF:M09763_1 | $2.15 \times 10^{-3}$ |
| CREB1; motif: NNNNSSGGCGCSSNNNNRTGACGTCAC; match class: 1 | TF:M12591_1 | $2.25 \times 10^{-3}$ |
| JunD; motif: RTGACGTCA | TF:M04681 | $3.30 \times 10^{-3}$ |
| ESR1; motif: STGACCTN | TF:M12602 | $3.70 \times 10^{-3}$ |
| Sp1; motif: NWRGCCACGCCCMCN; match class: 1 | TF:M12152_1 | $7.80 \times 10^{-3}$ |
| ER-alpha; motif: AGGTCASMNTGACCY | TF:M09909 | $1.04 \times 10^{-2}$ |

p300 or HDAC1 to NFκB and AP-1 binding sites promotes DNA loop formation [40]. GATA Zinc Finger Domain Containing 2A (GATAD2A) is a transcriptional repressor that enables protein-polymer adaptor activity. It is mainly responsible for chromatin compaction [41]. Sp3 Transcription Factor (SP3) is a transcription factor that regulates transcription by binding to GC and GT box regulatory elements. In human mammary carcinoma cells, SP3 binds to GC1 and GC2 elements of the topoisomerase IIα promoter, forming a DNA loop that can function either as a transcriptional activator or repressor [42]. Kruppel Like Factor 16 (KLF4) regulates transcription via RNA polymerase II. The expression of KLF4 is highly associated with stemness in human osteosarcoma carcinomas [43]. T-Box Transcription Factor 2 (TBX2) is the only T-box transcription factor that functions as a transcriptional repressor rather than a transcriptional activator. It is implicated in lung, breast, bone, pancreas, and melanoma cancers and represses transcription in human fetal kidney HEK293 cells by forming DNA loops in concert with HDAC and PBX1 [44]. Nuclear Factor I C (NFIC) is also known as CCAAT-Binding Transcription Factor. When bound to DNA, it invokes the core activator, which opens the chromatin structure to make room for the general transcription factor [32]. cAMP responsive element binding protein 1 (CREB1) cooperates with CTCF proteins to create complex protein-DNA interactions. It causes transcriptional repression and DNA loop formation in leukemic Jurkat T cells [45]. JunD Proto-Oncogene is a member of the JUN family and protects cells from apoptosis. In human hepatocytes, it binds to the CYP2C9 promoter and forms a DNA loop [46]. Estrogen Receptor 1 (ESR1) is associated with breast cancer, endometrial, and other types of cancer. Especially in normal breast epithelial cells, estrogen stimulation induces the formation of DNA loops in ESR1 at the 16p11.2 gene cluster [47]. Sp1 Transcription Factor (SP1) is involved in many processes, including cell differentiation, apoptosis, and chromatin remodeling. SP1 cooperates with the transcription factor GATA1 at erythroid-specific promoters in erythroid cells to form DNA loops close to distant enhancers [48].

**3.1.3 GSE149103 (pancreatic cancer).** Transcription factors related to DNA loop formation were enriched in three data sets. As a representative, the results of the enrichment analysis for Capan-1 cells are shown in Table 5.

LHM1 is a homeobox transcription factor that cooperates with the GATA protein in mice to facilitate the formation of long-range interactions [49]. OCT4 (POU class 5 homeobox 1)

**Table 5. Enrichment analysis of Capan-1 Hi-C data by g:Profiler.**

| Term_Name | Term_ID | Adjusted_p_value |
|---|---|---|
| HOXA7; motif: GYMATTAN | TF:M10690 | $2.04 \times 10^{-3}$ |
| FOXM1; motif: NAGASTGATTA | TF:M04611 | $2.18 \times 10^{-3}$ |
| GATA-3; motif: AGATAAGATCT | TF:M12193 | $3.56 \times 10^{-3}$ |
| HMGIY; motif: NNKKNAWTTTNYTNN | TF:M01010 | $3.65 \times 10^{-3}$ |
| SATB1; motif: NTTTAT; match class: 1 | TF:M03564_1 | $6.09 \times 10^{-3}$ |
| C/EBPgamma; motif: YTBATTTCARAAW | TF:M00622 | $1.31 \times 10^{-2}$ |
| LIM-1; motif: NYAATTAN | TF:M11000 | $1.50 \times 10^{-2}$ |
| IRF; motif: RAAANTGAAAN | TF:M00972 | $2.10 \times 10^{-2}$ |
| Oct-4; motif: ATTGWSWTGCWAAWN; match class: 1 | TF:M01124_1 | $2.84 \times 10^{-2}$ |

plays a vital role in embryonic development and stem cell pluripotency. OCT4 forms a cohesin-dependent enhancer-promoter loop in embryonic cells and trimethylates H3K4 at the SOX-17 locus to activate the SOX-17 promoter is activated [50].

**3.1.4 GSE160235 (rectal cancer).** Transcription factors related to DNA loop formation were enriched in all datasets. As a representative, the enrichment analysis results for TOP2A2B_bothcontrol_30 are shown in Table 6.

In breast cancer cells, PARP binds to the base lesion region of the MAR and is involved in the chromatin loop [51]. IPF1, also known as PDX1 (Pancreatic and Duodenal Homeobox 1), is an insulin-promoting factor. In pancreatic cells, Pdx1 and BETA2/NeuroD1 form a DNA loop in insulin activity [52]. FOXP3 is a master transcription factor for regulatory T cells (Treg) and cooperates with NFAT to form long-range chromatin interactions in mouse Treg cells [53].

**3.1.5 GSE167150 (breast cancer).** Transcription factors involved in DNA loop formation were enriched in all datasets. As a representative, the enrichment analysis results of BT549 are shown and mentioned in Table 7.

**3.1.6 GSE168470 (lymphoma).** Transcription factors related to DNA loop formation were enriched in all datasets. As a representative, the results of the enrichment analysis for Patient_1_merged_rep12 are shown in Table 8.

**Table 6. Enrichment analysis of TOP2A2B_bothcontrol_30 Hi-C data by g:Profiler.**

| Term_Name | Term_ID | Adjusted_p_value |
|---|---|---|
| SATB1; motif: NTTTAT; match class: 1 | TF:M03564_1 | $8.09 \times 10^{-23}$ |
| PARP; motif: TTTCYN; match class: 1 | TF:M02027_1 | $2.39 \times 10^{-20}$ |
| HOXA13; motif: ATAAMA; match class: 1 | TF:M01292_1 | $2.14 \times 10^{-15}$ |
| HOXA3; motif: NNNNRNTAATTARY; match class: 1 | TF:M01337_1 | $5.66 \times 10^{-15}$ |
| C/EBPgamma; motif: YTBATTTCARAAW | TF:M00622 | $1.48 \times 10^{-14}$ |
| ipf1; motif: CATTAR | TF:M01275 | $2.79 \times 10^{-14}$ |
| IRF-4; motif: GAAARTA | TF:M01883 | $1.86 \times 10^{-13}$ |
| HMGIY; motif: NNKKNAWTTTNYTNN | TF:M01010 | $2.47 \times 10^{-12}$ |
| NFATc2; motif: GGAAAA; match class: 1 | TF:M01281_1 | $1.08 \times 10^{-11}$ |
| NFATc3; motif: GGAAAA; match class: 1 | TF:M01886_1 | $1.08 \times 10^{-11}$ |
| TATA; motif: STATAAAWRNNNNNN | TF:M00252 | $5.20 \times 10^{-9}$ |
| FOXP3; motif: NNNVAAACANWD | TF:M07419 | $2.83 \times 10^{-8}$ |

**Table 7. Enrichment analysis of BT549 Hi-C data by g:Profiler.**

| Term_Name | Term_ID | Adjusted_p_value |
|---|---|---|
| SATB1; motif: NTTTAT; match class: 1 | TF:M03564_1 | $1.03 \times 10^{-11}$ |
| PARP; motif: TTTCYN | TF:M02027 | $2.78 \times 10^{-10}$ |
| FOXM1; motif: TRTTTATNN | TF:M08883 | $8.54 \times 10^{-8}$ |
| LIM-1; motif: NYAATTAN | TF:M11000 | $7.65 \times 10^{-7}$ |
| C/EBPgamma; motif: YTBATTTCARAAW | TF:M00622 | $1.83 \times 10^{-6}$ |
| HFH2; motif: NYWAYRTAAACA | TF:M11554 | $1.16 \times 10^{-5}$ |
| TBP; motif: MTATAAAARS; match class: 1 | TF:M10088_1 | $1.77 \times 10^{-5}$ |
| IRF-4; motif: GAAARTA | TF:M01883 | $8.75 \times 10^{-5}$ |
| GATA-3; motif: AGATAAGATCT | TF:M12193 | $1.42 \times 10^{-4}$ |
| TATA; motif: STATAAAWRNNNNNN | TF:M00252 | $1.70 \times 10^{-4}$ |

**Table 8. Enrichment analysis of Patient_1_merged_rep12 by g:Profiler.**

| Term_Name | Term_ID | Adjusted_p_value |
|---|---|---|
| HOXA7; motif: GYMATTAN; match class: 1 | TF:M10690_1 | $1.01 \times 10^{-13}$ |
| NFATc2; motif: GGAAAA; match class: 1 | TF:M01281_1 | $4.64 \times 10^{-9}$ |
| NFATc3; motif: GGAAAA; match class: 1 | TF:M01886_1 | $4.64 \times 10^{-9}$ |
| C/EBPgamma; motif: YTBATTTCARAAW | TF:M00622 | $6.50 \times 10^{-9}$ |
| SATB1; motif: NTTTAT; match class: 1 | TF:M03564_1 | $6.91 \times 10^{-9}$ |
| HoxA5; motif: NYMATTAN | TF:M10706 | $1.03 \times 10^{-7}$ |
| ipf1; motif: NVSTAATTAC | TF:M01235 | $3.35 \times 10^{-6}$ |
| TBP; motif: NGNNTATAAAA | TF:M03581 | $2.05 \times 10^{-5}$ |
| LIM-1; motif: NYAATTAN | TF:M11000 | $6.60 \times 10^{-5}$ |

**Table 9. Enrichment analysis of HEK_HiC_NUP_IDR_FS_A9_1.1 by g:Profiler.**

| Term_Name | Term_ID | Adjusted_p_value |
|---|---|---|
| HMGIY; motif: NNKKNAWTTTNYTNN; match class: 1 | TF:M01010_1 | $1.52 \times 10^{-5}$ |
| Oct-4; motif: ATTGWSWTGCWAAWN | TF:M01124 | $2.35 \times 10^{-5}$ |
| FOXP3; motif: NNNVAAACANWD | TF:M07419 | $3.94 \times 10^{-5}$ |
| SATB1; motif: NTTTAT; match class: 1 | TF:M03564_1 | $1.11 \times 10^{-4}$ |
| PARP; motif: TTTCYN; match class: 1 | TF:M02027_1 | $1.44 \times 10^{-4}$ |
| LIM-1; motif: NYAATTAN | TF:M11000 | $2.07 \times 10^{-4}$ |
| NFATc3; motif: GGAAAA; match class: 1 | TF:M01886_1 | $1.66 \times 10^{-2}$ |
| NFATc2; motif: GGAAAA; match class: 1 | TF:M01281_1 | $1.66 \times 10^{-2}$ |
| CEBPA; motif: ATTGCAYAAYN | TF:M07205 | $3.39 \times 10^{-2}$ |

HFH2, also known as FOXD3, is involved in pluripotent cell development. FOXD3 binds to enhancer sites and collects the SWISNF chromatin remodeling complex ATPase BRG1 and induces chromatin ribo ring [54].

**3.1.7 GSE143465 (kidney).** Transcription factors related to DNA loop formation were enriched throughout the data. As a representative, the results of the enrichment analysis for HEK_HiC_NUP_IDR_FS_A9_1.1 are shown in Table 9.

## 3.2 Consistency between DNA loops and peaks of ChIP-seq and ATAC-seq

ChIP-seq or ATAC-seq data were analyzed in conjunction with the Hi-C data, and the top 300 peaks were checked to determine whether they matched the enhancer-promoter regions in this study. In the present study, Fisher's exact probability test was performed to determine whether the enhancer-promoter regions in this study were significantly consistent with random selections. A comparison of the enhancer-promoter regions by GSM6061749_LIMA_ChIP_h3k27ac _THP1_WT_LPIF_S_0090_1.1.1_peaks and GSM6061774 (90 min Hi-C data for macrophages) is shown in Table 10.

In GSM6061774, 318,380,000 bp were selected as loop regions, of which 2,410,018 bp were partially matched to peaks of the ChIP-seq data. The expected value was determined by multiplying the proportion of the top 300 peak regions (peak regions divided by total genomic regions) by the length of the loop region. The p-value for Fisher's exact probability test was $2.2 \times 10^{-16}$, and the odds ratio was 39.33, indicating significant agreement. Other data were also in significant agreement (S1 File).

## 4 Discussion

A previous study revealed that the MDS-based method could reproduce DNA loops more prominently than existing studies [7]. Therefore, the results presented here (by substituting missing values with values of zero) were compared with those from the previous method. The previous method assigns an average for each genomic coordinate distance to the missing values. Thus, it could not express the marked difference between the coordinates of loop formation and non-loop formation. Surprisingly, the current method could significantly distinguish between DNA with or without loops. Chromosome distance plots based on the present method and those based on the previous method are shown in Figs 6 and 7.

DNA loop regions in the whole genome selected by the previous method, miniMDS [9], and the current method were $316150000[bp]/3186000000[bp] = 0.0992$, $62840000[bp]/3186000000[bp] = 0.0197$, and $145900000[bp]/3186000000[bp] = 0.0458$, respectively. The number of DNA loops was 1,822, 2,235, and 3,860, and the average lengths were 77,650 bp, 28,090 bp, and 82,520 bp. The results of the enrichment analysis using the previous method are summarized in Table 11.

The results for the proposed MDS-based method are shown in Table 3. Moreover, the transcription factors involved in DNA loop formation were not enriched by miniMDS. Consequently, they indicate that the number of enriched transcription factors obtained by the present method was higher compared with that of the previous method.

The current study presents certain limitations. First, MDS was used to reproduce the genome structure, which is computationally expensive if the resolution of Hi-C data is improved. Second, if the Hi-C data set contains time series data, time-independent genomic structures must be identified beforehand. Third, although bulk Hi-C data were analyzed, the

**Table 10. Consistency between DNA loops and ChIP-seq peaks.**

|  | Peaks [bp] | Not Peaks [bp] |
|---|---|---|
| DNA loop in this study | 2410018.00 | 315969982.00 |
| Expected Value | 61730.32 | 318318270.00 |

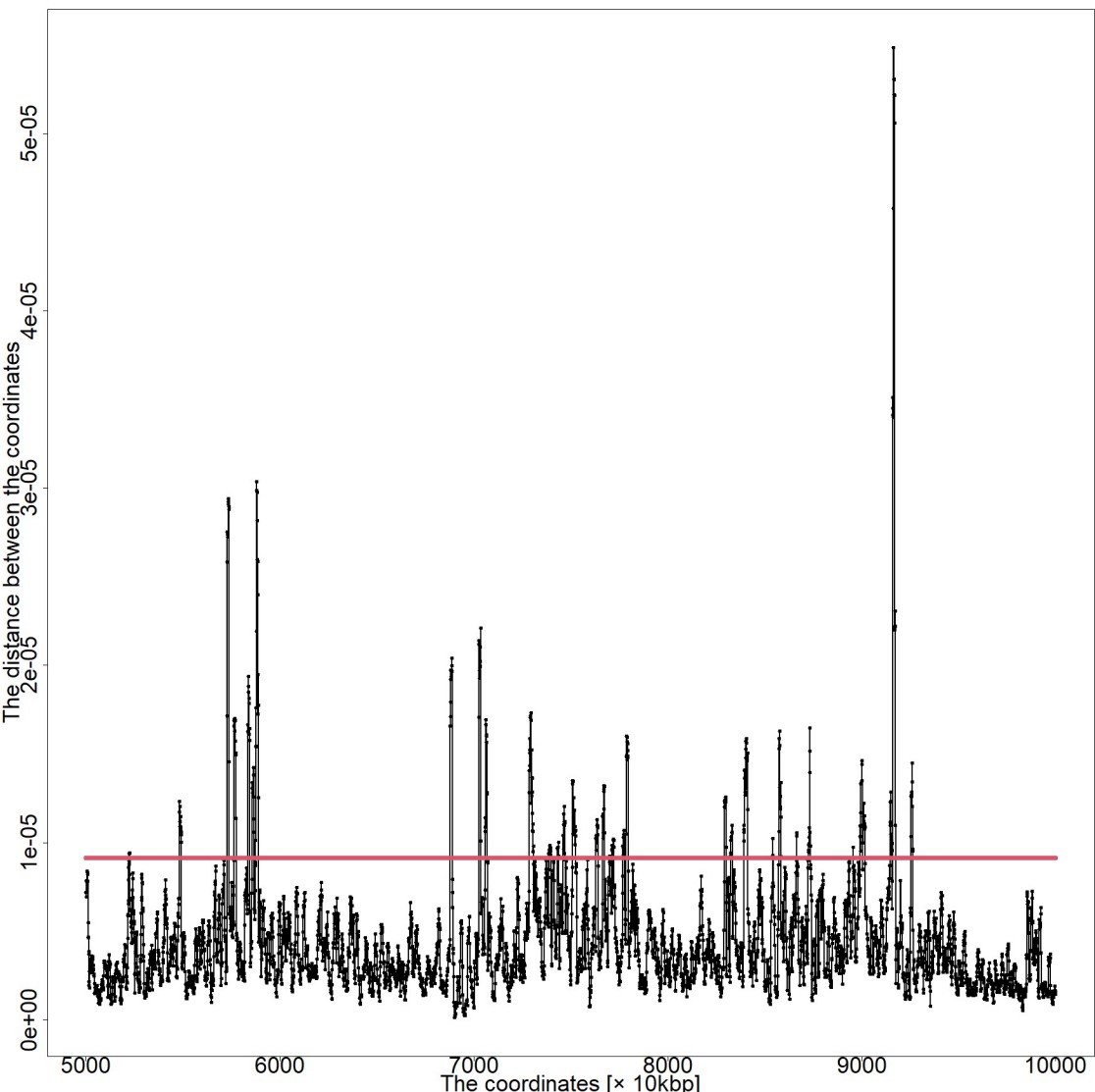

**Fig 6. Distance plot between coordinates of chromosome 5 50,000–100,000 kbp of series GSM6061774 by the MDS-based method.**

method should be applied to single cells to observe changes in genome structure, which presents a challenge for future studies.

The primary goal of this study was to identify a method to reconstruct genomic structures that differentiates DNA with or without loops. As indicated in the results, the validity of the proposed method is verified by three main findings: (i) The number of genes on DNA loop regions was large. (ii) Transcription factors involved in DNA loop formation were enriched in the enrichment analysis by g: profiler. (iii) The positions of the selected DNA loop regions and the ChIP-seq and ATAC-seq peaks were significantly consistent. To the best of my knowledge, no existing study has confirmed the consistency of the DNA loop to this extent. Therefore, the consistency of the three findings was confirmed in a data-driven manner, which is useful for reproducing DNA loops.

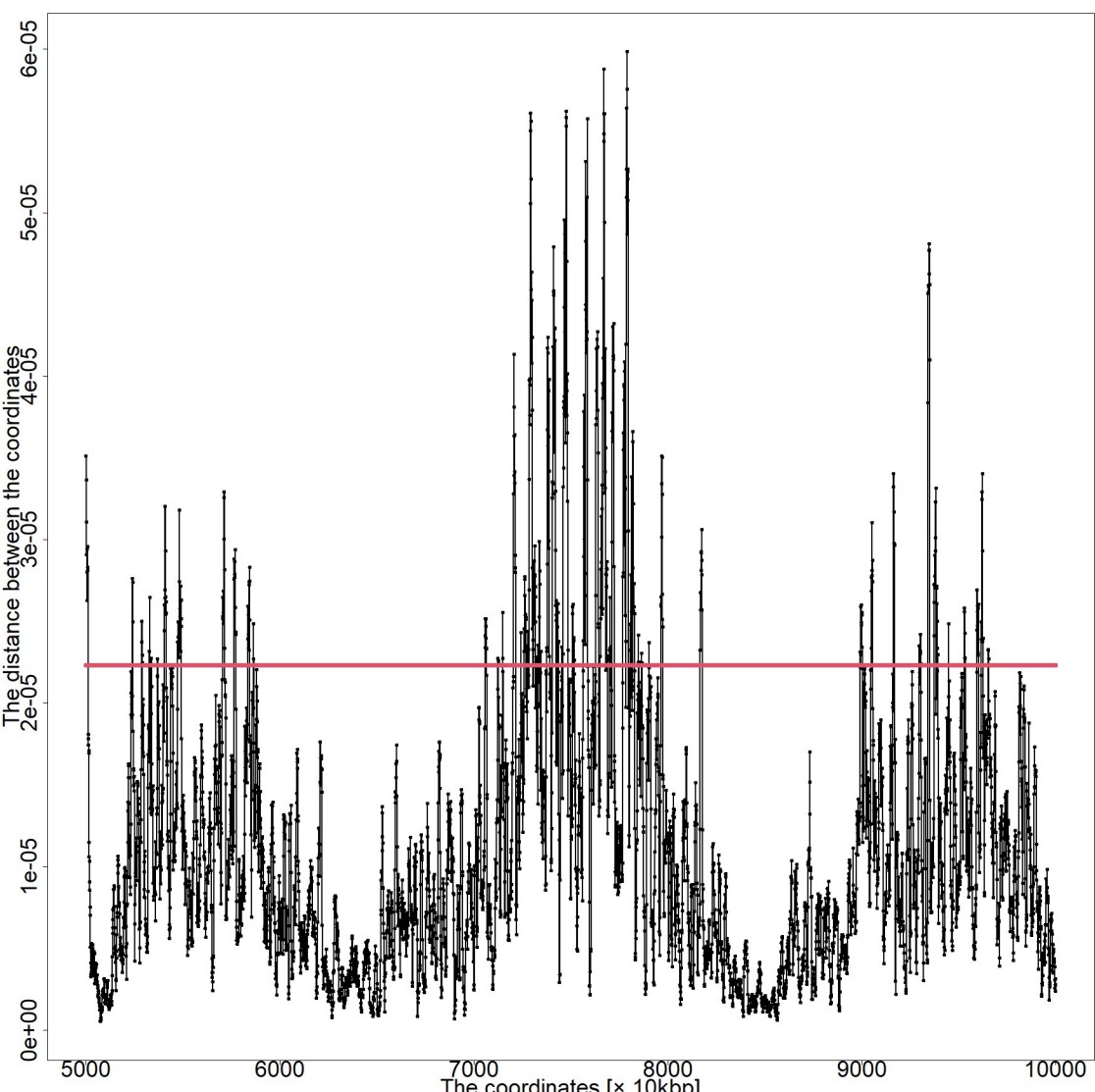

**Fig 7. Distance plot between coordinates of chromosome 5 50,000–100,000 kbp of series GSM6061774 by the previous MDS-based method.**

**Table 11. Enrichment analysis of Hi-C data collected over 90 min by the previous method.**

| Term_Name | Term_ID | Adjusted_p_value |
|---|---|---|
| IRF-4; motif: AAGTTTC; match class: 1 | TF:M04855_1 | 0.002042 |
| HOXA7; motif: GYMATTAN; match class: 1 | TF:M10690_1 | 0.004011 |
| ipf1; motif: NNRNTAATTAGYNCAN | TF:M01438 | 0.004204 |
| HoxA5; motif: NYMATTAN | TF:M10706 | 0.038841 |
| HOXA6; motif: NYMATTAN | TF:M08772 | 0.048191 |

## Supporting information

**S1 File.**
(ZIP)

## Acknowledgments

I am grateful to my supervisor Prof. Y-h. Taguchi for providing valuable discussions.

## Author Contributions

**Conceptualization:** Ryo Ishibashi.

**Data curation:** Ryo Ishibashi.

**Formal analysis:** Ryo Ishibashi.

**Investigation:** Ryo Ishibashi.

**Methodology:** Ryo Ishibashi.

**Project administration:** Ryo Ishibashi.

**Software:** Ryo Ishibashi.

**Supervision:** Ryo Ishibashi.

**Validation:** Ryo Ishibashi.

**Visualization:** Ryo Ishibashi.

**Writing – original draft:** Ryo Ishibashi.

**Writing – review & editing:** Ryo Ishibashi.

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
