## [Decision Letter · Decision Letter 0]

14 Feb 2023

PONE-D-22-35643Multidimensional Scaling Methods Can Reconstruct Genomic DNA Loops Using Hi-C Data PropertiesPLOS ONE

Dear Dr. Ishibashi,

Thank you for submitting your manuscript to PLOS ONE. After careful consideration, we feel that it has merit but does not fully meet PLOS ONE’s publication criteria as it currently stands. Therefore, we invite you to submit a revised version of the manuscript that addresses the points raised during the review process.

We look forward to receiving your revised manuscript.

Kind regards,

Divijendra Natha Reddy Sirigiri

Academic Editor

PLOS ONE

Journal Requirements:

Reviewers' comments:

Reviewer's Responses to Questions

**Comments to the Author**

1. Is the manuscript technically sound, and do the data support the conclusions?

Reviewer #1: Yes

Reviewer #2: Partly

Reviewer #3: Partly

2. Has the statistical analysis been performed appropriately and rigorously? 

Reviewer #1: Yes

Reviewer #2: No

Reviewer #3: No

3. Have the authors made all data underlying the findings in their manuscript fully available?

Reviewer #1: No

Reviewer #2: Yes

Reviewer #3: No

4. Is the manuscript presented in an intelligible fashion and written in standard English?

Reviewer #1: Yes

Reviewer #2: No

Reviewer #3: No

5. Review Comments to the Author

Reviewer #1: The author demonstrated an application of the multidimensional scaling (MDS) method to Hi-C data for visualizing DNA loops. In general, Hi-C data have information on population-averaged 3D chromatin structure, and the data format finally becomes a matrix of the contact frequency. First, the author transformed the contact matrix into a distance matrix. Then, the author tried to extract a 3D genome structural feature using the MDS method relating to the singular value decomposition. Next, he/she defined criteria to find DNA loops. Interestingly, his/her proposed method revealed that DNA loops form at active and open chromatin regions, which is consistent with a current viewpoint on the loop domain. Therefore, the reviewer found that the scientific treatments sound good, and these quantitative results would be reliable. On the other hand, the reviewer would like to require revisions based on the following comment:

1. In the main text, the author uses "ChIP-seq" and shows a table of the analysis of H3K27ac, which is related to the transcriptionally active state. However, the author did not mention the chromatin state or motivation to analyze ChIP-seq data. The author should mention such information.

2. How did the author find the logarithmic weight in Eq. (1)? What is the base of the logarithm? If the base is 5, the equations should make sense.

3. The base of the logarithm of the Y-axis in Figs. 1 and 2 should be described.

4. The reviewer could not understand the data format in Supplementary Files. For example, the file "GSE141067(osteosarcoma)/0min/SVD_234/plot.chr1.0-4980.txt" has 5 columns and 2989 rows.

5. l. 130 and Eq. (3): The author converted the contact frequency into the distance without any references. Although the assumption is not theoretically supported in terms of polymer physics [Shinkai et al. NARGAB 2020], some computational methods have conventionally assumed [Serra et al. FEBS Lett 2015].

6. Fig. 3 legend must be incomplete. The author only plotted the eigenvectors. Moreover, the figure includes blue lines without explanation.

7. What does the author mean by "roots" in Fig. 4 legend?

8. Eq. (7): The variable "V" seems to be a new one. What is the difference between "v" and "V"?

9. l. 174-183: Can this algorithm uniquely determine loop regions? This definition must be the most critical part of this work. However, the reviewer could not understand it and Fig. 4 correctly. A schematic of the algorithm would help broad readers.

10. p. 6: Please remove the Fig.5 legend.

11. l. 207: The author describes, "The number of DNA loops and number of genes are close to each other," referring to Table 1. However, the number of DNA loops is almost 30000, and the number of genes is almost 5000 in Table 1. These are NOT close to each other.

12. §3.2: Has the author analyzed H3K27me3 ChIP-seq data? The histone marker is related to heterochromatin.

13. Figure 7 is not attached to the manuscript.

Reviewer #2: In this manuscript, Ryo utilizes multidimensional scaling (MDS) to capture DNA loops for high-throughput chromosome conformation capture (HiC) data. By transforming HiC data to a distance matrix, MDS identified DNA loops contained more DNA bound by transcription factors than existing methods. The manuscript is well organized.

I have 4 major concerns.

1. In section 2.2.2, the term “exaggerated representations of DNA loops” is confusing. Could you clarify the definition and rationale behind it? I am also concerned about the threshold of 50kbp for formula 1. Would the significant loops of length 50kbp be affected by this transformation?

2. In section 2.2.3, the authors claimed that the second and the third eigenvectors often retain the original structure. Empirically, how would one decide for real HiC data which eigenvectors to use for identifying loops? Why is the first eigenvector excluded?

3. In formula 8, the distance matrix is smoothed with a window size of 100kbp. Has the author evaluated the performance of MDS varying different window sizes? For other HiC data, how would the user recommend selecting this hyperparameter?

4. In table 1, is 318380 the number of all potential DNA loops considered? What do the ratios in two columns represent? # of loops seems much bigger than # of genes in table 1, which contradicts the statement in lines 207-210, “the number of loops and number of genes are close to each other?” Could you quantify this?

I have 3 minor comments.

1. It would be helpful to have a table of datasets to summarize the read depth, bin size, year, and HiC data type.

2. Below line 195, the description of figure 5 should be revised.

3. The grammar and wording can be much improved for this manuscript.

Reviewer #3: Ishibashi suggested a method (MDS) to predict DNA loops based on Hi-C sequencing data. It tested on 7 publicly available Hi-C datasets from various cancers (e.g., breast cancer, lymphoma, and kidney cancer). However, there are four major problems in the manuscript:

1) In the Methods section, the description of MDS in predicting DNA loops is not so clear. The algorithm, code and demo data for this study shall be publicly available for others, in order to reproduce the results and understand the method. The prediction accuracy for DNA-loops has to be rigorously evaluated by proper statistical methods.

2) In the Results section, the presentation of predicted loops in each dataset is too short to evaluate its merit. In Particular, the results have to be compared with other similar tools for predicting DNA loops by using Hi-C sequencing datasets. And also, I am not clear about how other genomic information (e.g., enhancer or promoter histone markers and nucleosome density) at these predicted loops, which are important markers for validating the predicted DNA loops/interactions, are functional or random interactions.

3) In the Discussions section, the author needs to improve this part significantly to convince people that the proposed new method is useful for studying DNA loops based on Hi-C data.

4) Finally, the quality of all main figures is not good, which has to be improved significantly.

6. PLOS authors have the option to publish the peer review history of their article (what does this mean?). If published, this will include your full peer review and any attached files.

Reviewer #1: No

Reviewer #2: No

Reviewer #3: No

---

## [Author Response · Author response to Decision Letter 0]

18 Mar 2023

Responses to the reviewer were presented in PDF format.

---

## [Decision Letter · Decision Letter 1]

28 Apr 2023

PONE-D-22-35643R1Multidimensional Scaling Methods Can Reconstruct Genomic DNA Loops Using Hi-C Data PropertiesPLOS ONE

Dear Dr. Ishibashi,

Thank you for submitting your manuscript to PLOS ONE. After careful consideration, we feel that it has merit but does not fully meet PLOS ONE’s publication criteria as it currently stands. Therefore, we invite you to submit a revised version of the manuscript that addresses the points raised during the review process.

There are minor issues to be further addressed. please submit the revised manuscript

We look forward to receiving your revised manuscript.

Kind regards,

Divijendra Natha Reddy Sirigiri

Academic Editor

PLOS ONE

Journal Requirements:

Reviewers' comments:

Reviewer's Responses to Questions

**Comments to the Author**

1. If the authors have adequately addressed your comments raised in a previous round of review and you feel that this manuscript is now acceptable for publication, you may indicate that here to bypass the “Comments to the Author” section, enter your conflict of interest statement in the “Confidential to Editor” section, and submit your "Accept" recommendation.

Reviewer #1: All comments have been addressed

2. Is the manuscript technically sound, and do the data support the conclusions?

Reviewer #1: Yes

3. Has the statistical analysis been performed appropriately and rigorously? 

Reviewer #1: Yes

4. Have the authors made all data underlying the findings in their manuscript fully available?

Reviewer #1: Yes

5. Is the manuscript presented in an intelligible fashion and written in standard English?

Reviewer #1: No

6. Review Comments to the Author

Reviewer #1: The author revised all concerns of the reviewer. However, there are still points that need to be clarified according to the following comments:

Major points:

1. Equations (4) -- (6).

Although the author describes a general framework for the eigenvalue decomposition or the singular value decomposition, it is still unclear how to convert the matrix D into the vector X (= (x_1, x_2, ..., x_N) ?). Furthermore, the definition of the second and the third eigenvectors are insufficient. The order of the corresponding eigenvalues is not defined.

2. p6. l175.

The author should understand that E_i does NOT have the physical unit because the matrix V is a unitary matrix, and the eigenvectors are normalized without the physical unit. Therefore, E_i is NOT the "physical" distance.

3. p7. ll191--196.

The definition and algorithm of the root were improved and readable. More simply put, is the location of the local minimum of E^-_i the root?

Typos:

1. p2. l23.

stoHi-C -> The stoHi-C

2. p4. l105.

Fig 1 shows -> Figure 1 shows

7. PLOS authors have the option to publish the peer review history of their article (what does this mean?). If published, this will include your full peer review and any attached files.

Reviewer #1: No

---

## [Author Response · Author response to Decision Letter 1]

6 May 2023

Comments to reviewers are shown in Response to Reviewers.pdf. 

This paper was proofread by Editage and written in standard English.

---

## [Editor Report · Decision Letter 2]

24 Jul 2023

Multidimensional Scaling Methods Can Reconstruct Genomic DNA Loops Using Hi-C Data Properties

PONE-D-22-35643R2

Dear Dr. Ishibashi,

We’re pleased to inform you that your manuscript has been judged scientifically suitable for publication and will be formally accepted for publication once it meets all outstanding technical requirements.

Kind regards,

Divijendra Natha Reddy Sirigiri

Academic Editor

PLOS ONE
---

## [Editor Report · Acceptance letter]

8 Aug 2023

PONE-D-22-35643R2 

Multidimensional Scaling Methods Can Reconstruct Genomic DNA Loops Using Hi-C Data Properties 

Dear Dr. Ishibashi:

I'm pleased to inform you that your manuscript has been deemed suitable for publication in PLOS ONE. Congratulations! Your manuscript is now with our production department. 

Kind regards, 

on behalf of

Dr. Divijendra Natha Reddy Sirigiri 

Academic Editor

PLOS ONE